# Follow-up of antibody changes in brucellosis patients in Gansu, China

Hanyu Sha,[1] Qun Duan,[1] Dongyue Lyu,[1] Fang Qian,[2] Xiaojin Zheng,[3] Jiazhen Guo,[2] Zhaokai He,[4] Xinmin Lu,[3] Asaiti Bukai,[3] Shuai Qin,[1] Ran Duan,[1] Saier Guli,[3] Peng Zhang,[5] Meng Xiao,[1] Huaiqi Jing,[1] Xin Wang[1]

**ABSTRACT** Brucellosis, caused by the intracellular bacterium *Brucella*, often leads to chronic infection with recurrent symptoms, posing significant therapeutic challenges. Effective monitoring of antibody dynamics is crucial for understanding disease progression and enhancing clinical outcomes. To investigate the longitudinal dynamics of antibody titers and antigen-specific responses in animal husbandry practitioners exposed to brucellosis in Gansu, China. We conducted a serological study involving 400 animal husbandry workers, identifying 21 seropositive individuals (5.25%). Antibody titers and antigen-specific responses to lipopolysaccharide (LPS) and BP26 protein were assessed using the Rose-Bengal test (RBT), serum agglutination test (SAT), and western blotting. Acute brucellosis patients displayed a distinct three-phase antibody titer pattern: a rapid rise (12–38 days), a rapid decline (19–41 days), and a slow decline (42–148 days). BP26-specific antibodies persisted significantly longer than LPS antibodies, remaining detectable up to 395 days post-infection. Early treatment reduced acute-phase antibody titers, correlating with disease control. This study combined dynamic antibody titer monitoring with specific antigen monitoring, enhancing the accuracy and specificity of brucellosis diagnosis and offering a basis for chronic infection detection. The findings are highly significant for clinical practice and public health strategies, particularly in high-incidence areas, prevention and control, as well as high-risk population surveillance.

**IMPORTANCE** *Brucella*, a zoonotic intracellular bacterium, poses significant threats to both human health and economic security. Clinically, brucellosis can be challenging to differentiate from other long-term febrile illnesses, necessitating prompt and standardized treatment to prevent chronic persistent infections and multi-organ damage, which are notoriously difficult to treat. Thus, gaining a comprehensive understanding of the disease's progression is essential for early diagnosis and effective treatment strategies. This paper delves into the dynamics of serum antibody titers in patients with acute brucellosis, shedding light on the temporal patterns of antibody titers. Such insights are pivotal for monitoring disease progression and assessing the efficacy of treatment interventions. Furthermore, through western blotting analysis, the study reveals that antibodies against the BP26 protein in brucellosis patients persist over an extended period, which is helpful to identify the durable immune response of brucellosis and provides a theoretical basis for vaccine development and treatment strategy formulation.

**KEYWORDS** brucellosis, zoonotic diseases, antibody titer, serology, BP26 protein, lipopolysaccharide

Brucellosis, a zoonotic disease caused by *Brucella* species, remains a global public health challenge, primarily in regions reliant on livestock economies (1). Human transmission occurs primarily through direct contact with infected animals

**Peer Reviewer** Mustafa Kürşat Şahin, Ondokuz Mayıs University, Atakum, Turkey

Address correspondence to Xin Wang, wangxin@icdc.cn.

Hanyu Sha, Qun Duan, Dongyue Lyu, Fang Qian, Xiaojin Zheng, Jiazhen Guo, and Zhaokai He contributed equally to this article. Author order was determined by drawing straws.

The authors declare no conflict of interest.

or consumption of unpasteurized dairy products (2). The pathogen exhibits unique virulence mechanisms, including intracellular survival strategies and immune evasion capabilities (3, 4), which complicate differential diagnosis, therapeutic management, and prognostic evaluation. Clinical manifestations are nonspecific and highly variable, frequently leading to diagnostic delays or misclassification (5–7). Untreated acute brucellosis often progresses to chronic infection, resulting in severe multi-organ complications such as spondylitis, arthritis, endocarditis, and neurological impairment (8–11). Conversely, early initiation of standardized antimicrobial therapy significantly reduces disease duration, symptom severity, and risks of chronicity or relapse (12).

Current brucellosis diagnostics rely on microbiological culture and serological assays (13). Though bacterial culture is the gold standard, its clinical application is limited by demanding culture conditions and biosafety risks. Thus, serological methods are preferred for routine diagnosis. Antibody dynamic analysis is central to immune monitoring, revealing the temporal characteristics and clinical significance of humoral immunity. Its quantitative parameters can serve as biomarkers for diagnosis, therapeutic efficacy evaluation, and prognostic stratification. A decline in antibody titers during treatment indicates pathogen clearance, whereas stable or rising titers may suggest suboptimal therapeutic response or drug resistance, warranting reevaluation. Persistent elevation of antibodies after treatment for chronic brucellosis indicates bacterial residual presence, necessitating adjustments to the treatment regimen (14). Population-based serological monitoring can track transmission patterns, assess intervention effectiveness, and guide prevention and control strategies (15).

Lipopolysaccharide (LPS) is a key component of the outer membrane and serves as an important virulence factor for *Brucella*. It plays a crucial role in the evasion of the innate immune response, allowing *Brucella* to survive, replicate, and establish chronic infections within host cells (16, 17). In the serological diagnosis of brucellosis, LPS is the primary antigen and exhibits cross-reactivity with *Yersinia enterocolitica O:9* due to shared antigens (18, 19), which undermines its diagnostic reliability in brucellosis. BP26 protein is highly conserved in Brucella and can induce a protective immune response with high immunogenicity (20) and higher specificity, offering a promising complementary biomarker (21–26). Yet, longitudinal data on BP26 antibody dynamics, especially in high-burden regions, remain limited.

This study focused on Akesai Kazakh Autonomous County, Gansu Province, a pastoral region within the China Qinghai-Tibet Plateau. Building on prior seroepidemiological surveys (6, 27), we conducted a longitudinal analysis of 21 acute brucellosis patients. We employed a multimodal serological protocol combining the Rose-Bengal test (RBT) for rapid screening, serum agglutination test (SAT) for quantitative antibody titration, and western blotting to characterize immune responses against key *Brucella* antigens—LPS and BP26 protein (28–30). We aimed to provide in-depth insights into the humoral immune dynamics of brucellosis and valuable recommendations for its serological diagnosis, treatment strategies, and surveillance.

## RESULTS

### Prevalence of brucellosis in China

In China, the prevalence of human brucellosis cases showed an overall increase from 2021 to 2024 (31), with the number of cases reaching as high as 72,344 in 2023. The annual prevalence trend of brucellosis in Chinese cases from 2010 to 2024 is shown in Fig. 1.

### Basic information on the study subjects

In this study, animal husbandry practitioners were recruited in Akesai Kazakh Autonomous County, located in the *Marmota himalayana* plague, focusing on the Altun Mountains of the Qinghai–Tibet Plateau. These practitioners raise sheep, camels, and other animals such as cattle and horses, which serve as the main *Brucella* reservoirs.

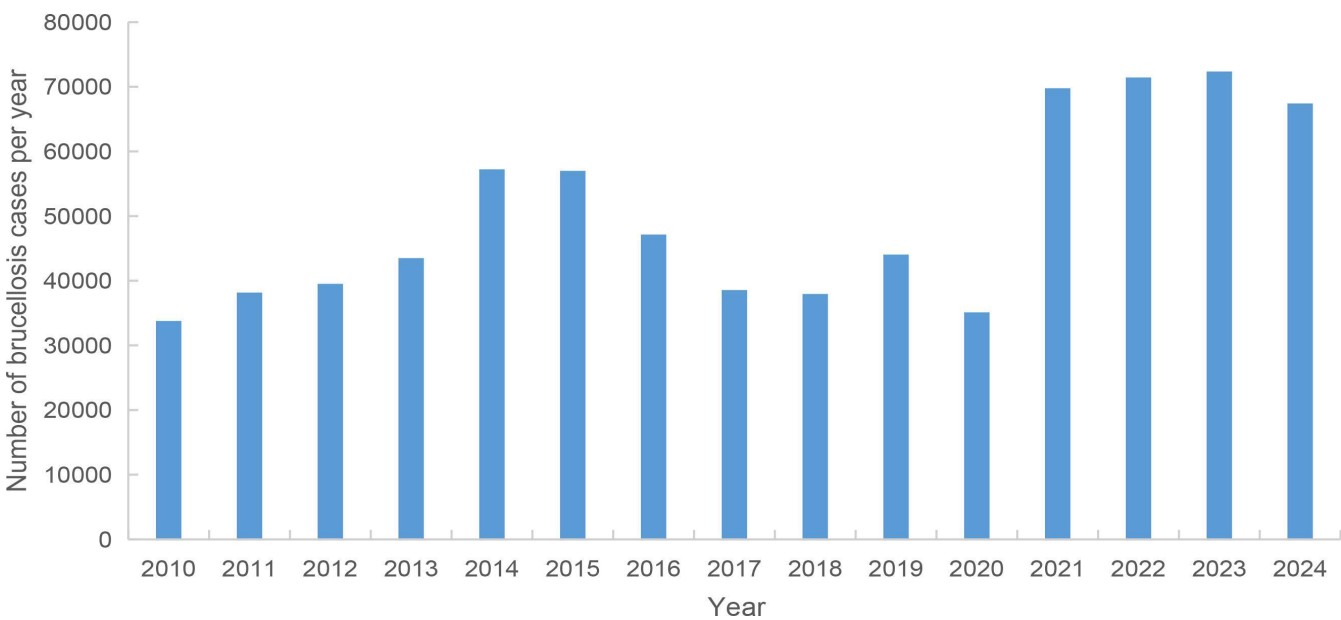

**FIG 1** Prevalence of brucellosis among patients in China from 2010 to 2024 (annual trends in the number of cases).

People can be infected with *Brucella* via contact with infected livestock products/ substances such as milk, meat, pus, urine, vaginal secretions, and even fecal aerosols. A total of 400 animal husbandry practitioners were included. Among these, 21 tested were positive for antibodies using RBT and SAT, resulting in a positivity rate of 5.25% (21/400) (detailed case information can be found in Table S1). Among the 21 antibody-positive patients, 19 were in the acute phase and 2 were in the chronic phase. Male patients accounted for the vast majority (90.48%; 19/21). The positive individuals had direct contact with animals, and the majority of them were mainly farmers (involved in breeding) (66.67%; 14/21), followed by herdsmen (28.57%; 6/21), and lastly veterinarians (4.76%; 1/21). Passive monitoring identified 71.43% (15/21) of the cases, which comprised all the cases with symptoms before blood collection, primarily fever (71.43%; 15/21) and fatigue (66.67%; 14/21). The longest interval between symptom onset and blood collection was 49 days, while the shortest was 6 days. Active surveillance detected 28.57% (6/21) of the cases, which comprised all the cases without symptoms; these six patients received treatment upon testing positive for antibodies.

## Changes in serum antibody titers in brucellosis patients

Regarding the 19 acute-phase patients, 17 exhibited changes in antibody titers (Fig. 2). The process from symptom onset to the disappearance of serum antibodies could be roughly divided into three periods: rapid rise, rapid decline, and slow decline of antibody titers. (i) There were three patients (AKS05, AKS06, and AKS07) in the rapid rise period. The antibody titers peaked 12–38 days after the first blood sample collection. During the observation period, the antibody titers reached 1:400++ in 7 cases, 1:800++ in 3 cases, and 1:1,600++ in 2 cases. (ii) There were six patients (AKS01, AKS03, AKS04, AKS05, AKS06, and AKS09) in the rapid decline period when a high antibody titer (>1:200++) fell to a relatively low titer (1:100++/1:200++) over 19–41 days. (iii) After the antibody titer decreased to 1:100++/1:200++, the rate of decline of the antibody titer became slow. There were eight patients (AKS01, AKS03, AKS04, AKS05, AKS12, AKS02, AKS06, and AKS09) in the slow decline period, of which five patients (AKS01, AKS03, AKS04, AKS05, and AKS12) experienced 42–148 days before the antibody titers decreased from 1:100++/1:200++ to <1:50++ (SAT negative). In addition, patient AKS02 experienced a decrease from 1:200++ to 1:100++ over 69 days, patient AKS06 experienced a decrease from 1:200++ to 1:50++ over 69 days, and patient AKS09 experienced maintenance at

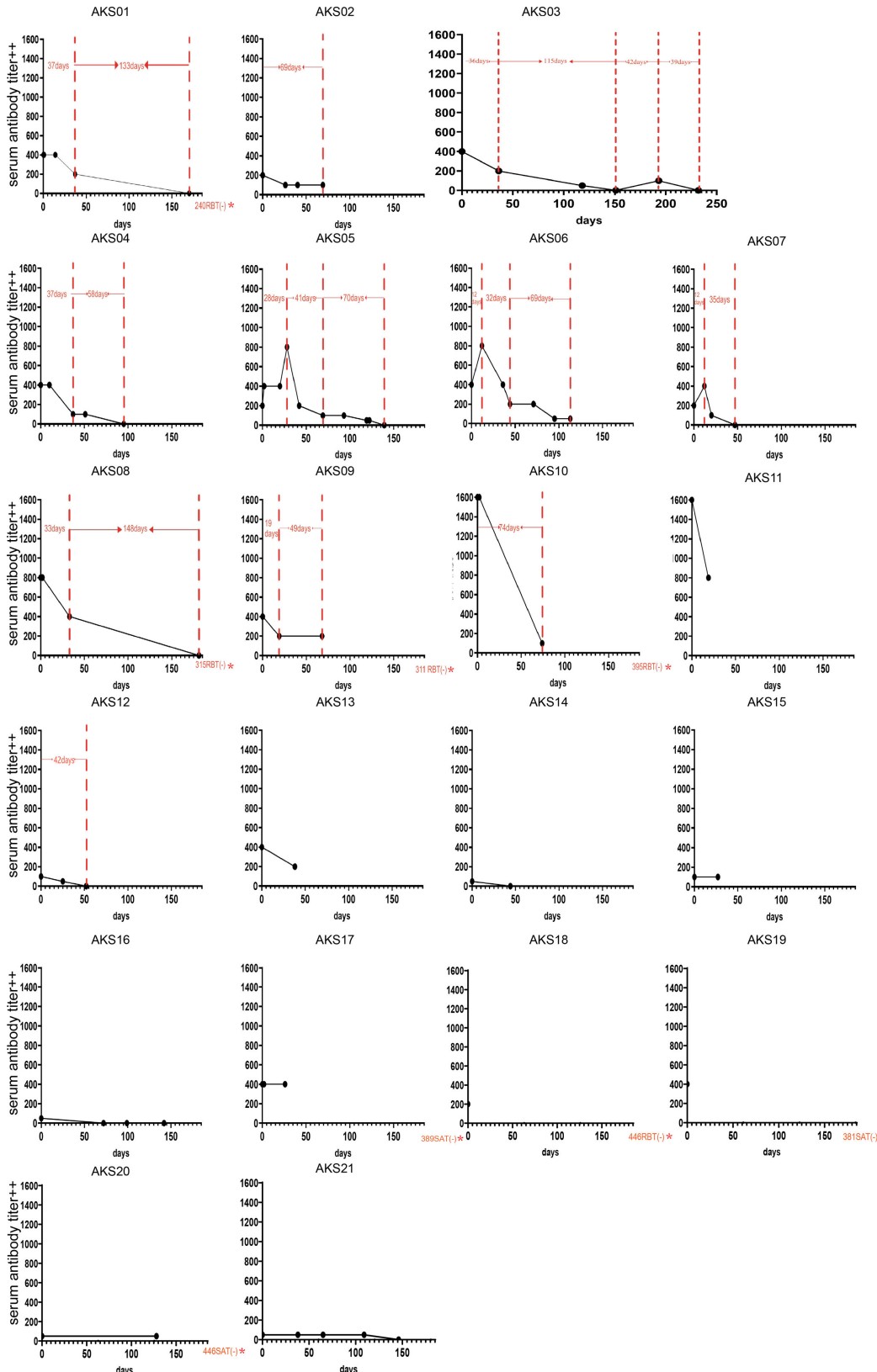

**FIG 2** Changes of serum antibody titers in patients with acute brucellosis (* indicates that the RBT/SAT test result of the patient's serum was negative from the first blood collection up to the specified number of days).

1:200++ for 49 days. Unusually, the AKS03 patient's antibody titer of 1:400++ at the first blood sample collection decreased to <1:50++ (SAT negative) over 151 days; after 42 days of SAT negativity, it increased to 1:100++, and after 39 days, it had returned to <1:50++ (SAT negative).

Regarding the two chronic patients (AKS20 and AKS21), the antibody titers were maintained at 1:50++ for 4–5 months. Serum antibody titers decreased or even disappeared in two chronic patients, and there was no objective evidence of infection, such as fever; however, self-reported fatigue-like symptoms persisted.

Regarding the changes in antibody titers (over 7 days) in the four patients in a Beijing hospital, three patients exhibited no changes (their antibody titers were maintained at 1:400++, 1:200++, and 1:100++, respectively), while the antibody titer of the remaining patient decreased from 1:800++ to 1:400++.

## Western blotting results

The target band of hybridization between *Brucella* antibody-positive serum and BP26 protein was mainly located at 45.3 kDa, while the target band of hybridization with LPS was mainly located at 30 kDa (Fig. 3). The western blotting results of the hybridization of serum of brucellosis patients (with different antibody titers) with BP26 protein and LPS are shown in Table 1. The results of serum hybridization with BP26 protein were positive in all patients. The results of serum hybridization with LPS showed that two patients (AKS07 and AKS12) had negative hybridization with LPS. The gray intensity of the bands hybridized with the BP26 protein in the serum of these two patients (AKS07 and AKS12) was not high. The results of serum hybridization with LPS showed that the higher the serum antibody titer in a particular patient, the higher the band grayscale intensity. Western blotting results of the hybridization reaction between serum and BP26, as well as LPS from each patient, are presented in Fig. 4. Annotations link the bands to the corresponding antibody titers and specific patient samples. Due to logistical constraints, the western blotting results corresponding to antibody titers in these patients were missing: 1:400 for AKS05, 1:400 and RBT− for AKS08, 1:800 for AKS11, 1:50 for AKS14, 1:200 and RBT− for AKS18, 1:50 for AKS20, and 1:50 for AKS21.

## DISCUSSION

Brucellosis persists as a critical global health threat, with escalating incidence rates observed across multiple regions worldwide, particularly in livestock-intensive developing nations with insufficient zoonotic disease control (32). Serological monitoring in Aksai Kazak Autonomous County, Gansu Province, among livestock practitioners showed brucellosis is prevalent, indicating a significant occupational exposure risk. Acute brucellosis patients displayed a distinct three-phase antibody titer pattern: a rapid rise, a rapid decline, and a slow decline. Western blot analysis revealed the persistence of antibodies targeting the BP26 protein, which may serve as a useful marker for disease monitoring. These findings provide an immunological basis for optimizing diagnosis and treatment routes and monitoring strategies.

The cases of brucellosis have been on the rise in China in recent years. The serum antibody positive rate of 5.25% (21/400) among livestock practitioners in our study aligns with that in other high-prevalence areas like Mongolia (33), the Mediterranean, and Middle Eastern countries(34). The untreated asymptomatic brucellosis carries significant risks of progressing to chronic complications. The detection of asymptomatic infections in this study and previous reports of numerous asymptomatic infections in high-risk populations from other regions highlights the crucial role of active surveillance (35). This study showed that the antibody titers of patients with acute brucellosis were 19 to 41 days of rapid decline to below 1:200++. Similarly, one study reported a significant decrease in SAT positivity and antibody titers (≥1:200++) at 42 days (6 weeks) post-treatment in acute brucellosis patients (12). The previous serological studies' conclusion was that timely antibacterial therapy can effectively suppress pathogen activity, leading to a rapid post-treatment decline in elevated titers (27, 31). This antibody kinetic

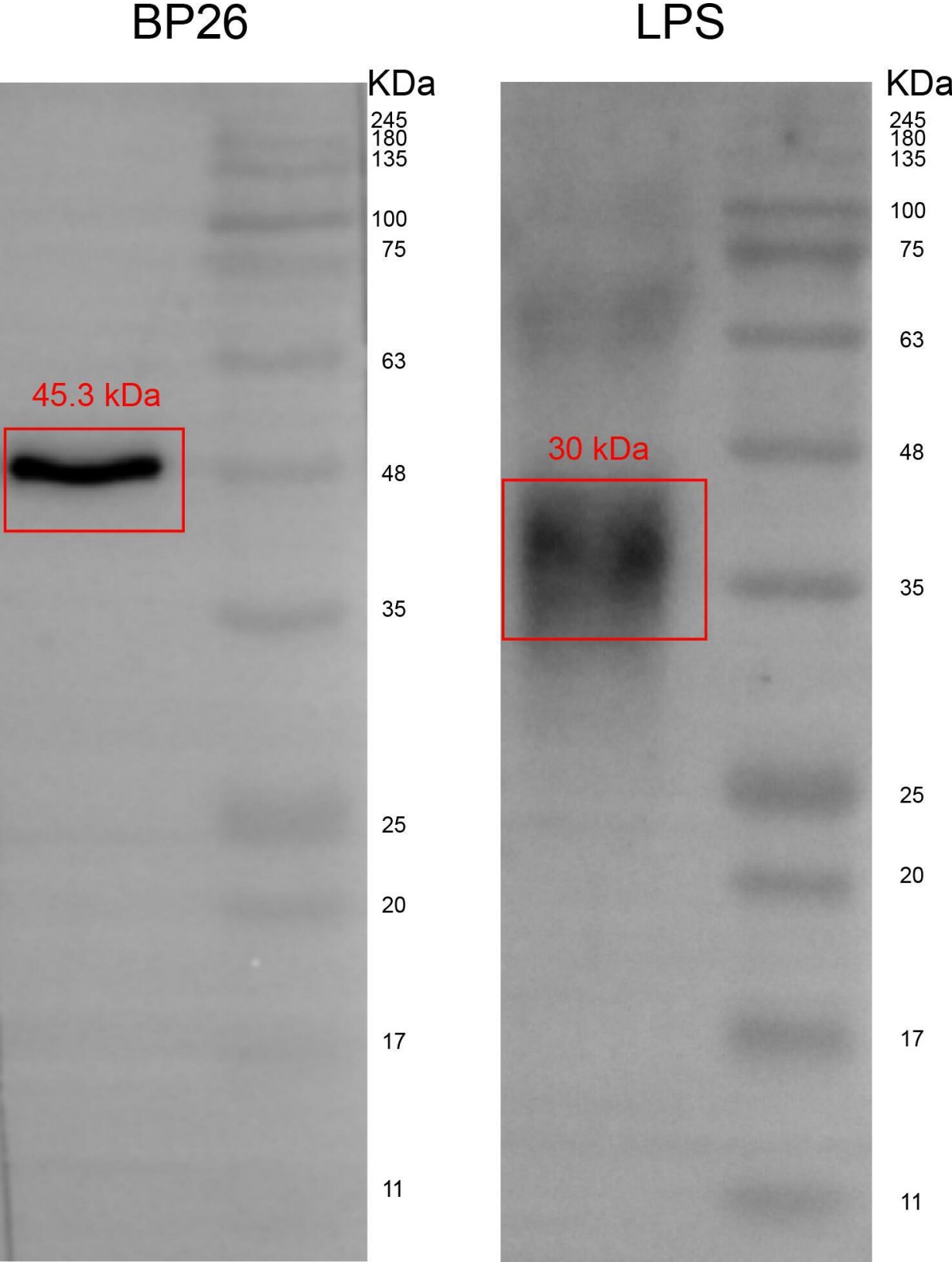

**FIG 3** Western blot analysis showing the reactivity of antibodies against BP26 protein and LPS (the molecular weight markers are indicated in kDa).

**TABLE 1** Results of western blot hybridization of serum with BP26 and LPS[a]

| Antibody titers | RBT(−) | | <1:50++ | | 1:50++ | | 1:100++ | | 1:200++ | | 1:400++ | | 1:800++ | | 1:1,600++ | |
|---|---|---|---|---|---|---|---|---|---|---|---|---|---|---|---|---|
| Antigen | BP26 | LPS | BP26 | LPS | BP26 | LPS | BP26 | LPS | BP26 | LPS | BP26 | LPS | BP26 | LPS | BP26 | LPS |
| AKS01 | + | + | + | + | / | / | / | / | + | + | + | + | / | / | / | / |
| AKS02 | / | / | / | / | / | / | + | + | + | + | / | / | / | / | / | / |
| AKS03 | / | / | + | + | + | + | + | + | + | + | + | + | / | / | / | / |
| AKS04 | / | / | + | + | / | / | + | + | / | / | + | + | / | / | / | / |
| AKS05 | / | / | + | + | + | + | + | + | + | + | / | / | + | + | / | / |
| AKS06 | / | / | / | / | + | + | / | / | + | + | + | + | + | + | / | / |
| AKS07 | / | / | + | − | / | / | + | − | + | − | + | − | / | / | / | / |
| AKS08 | / | / | + | + | / | / | / | / | / | / | / | / | + | + | / | / |
| AKS09 | + | + | / | / | / | / | / | / | + | + | + | + | / | / | / | / |
| AKS10 | + | ± | / | / | / | / | + | + | / | / | / | / | / | / | + | + |
| AKS11 | / | / | / | / | / | / | / | / | / | / | / | / | / | / | + | + |
| AKS12 | / | / | + | − | + | − | + | ± | / | / | / | / | / | / | / | / |
| AKS13 | / | / | / | / | / | / | / | / | + | + | + | + | / | / | / | / |
| AKS14 | / | / | + | + | / | / | / | / | / | / | / | / | / | / | / | / |
| AKS15 | / | / | / | / | / | / | + | + | / | / | / | / | / | / | / | / |
| AKS16 | + | ± | + | + | + | + | / | / | / | / | / | / | / | / | / | / |
| AKS17 | / | / | + | + | / | / | / | / | / | / | + | + | / | / | / | / |
| AKS18 | / | / | / | / | / | / | / | / | / | / | / | / | / | / | / | / |
| AKS19 | / | / | ± | + | / | / | / | / | / | / | + | + | / | / | / | / |
| AKS20 | / | / | + | ± | / | / | / | / | / | / | / | / | / | / | / | / |
| AKS21 | / | / | + | ± | / | / | / | / | / | / | / | / | / | / | / | / |

[a]+ indicates strongly positive, − indicates negative, ± indicates weakly positive, / indicates not available.

profile underscores the pivotal role of early therapeutic intervention in curbing disease progression. The antibody titers of two chronic brucellosis cases (patients AKS20 and AKS21) remained stable at 1:50++ for 4–5 months, with fatigue being the predominant clinical symptom. A previous case report from Japan described a chronic brucellosis patient who maintained an antibody titer of 1:20 for up to 8 months (36). These results indicate that the antibody titers of chronic brucellosis patients can be maintained at a low level for a long time. This is because *Brucella* can survive in cells for a long time, potentially by suppressing immunity, autophagy, and apoptosis through various mechanisms (37). Notably, one case (patient AKS03) exhibited fluctuating antibody titers, which warrants further investigation to determine whether this represented reinfection or relapse of chronic brucellosis. This phenomenon may be associated with the unique immunological features of *Brucella* infection (38). Therefore, early diagnosis and standardized treatment of acute brucellosis are crucial to prevent the progression of chronic disease.

In this study, western blot analysis revealed that BP26 antibodies persist longer than LPS antibodies, remaining detectable in 15 SAT-negative patients. This suggests that the BP26 protein can be a complementary diagnostic marker, particularly for chronic infections. Its slower decay rate extends the detection window, allowing reliable infection identification even when antibody titers fall below conventional thresholds. BP26 protein, with high immunogenicity and specificity, can induce immune responses and antibody production in the early stage of infection (39). Animals vaccinated with the BP26-based recombinant vaccine develop BP26 antibodies, but their immune response, in terms of antibody quantity and subtype, differs from that of naturally infected animals (40). Thus, the BP26 protein shows great potential in differentiating vaccination from natural infection, early diagnosis, detection of chronic brucellosis patients, and vaccine development (41).

Two SAT-positive patients (AKS07 and AKS12) tested negative for anti-LPS antibodies. This underscores the importance of considering potential confounding factors when analyzing western blot results to ensure the accuracy and reliability of the findings.

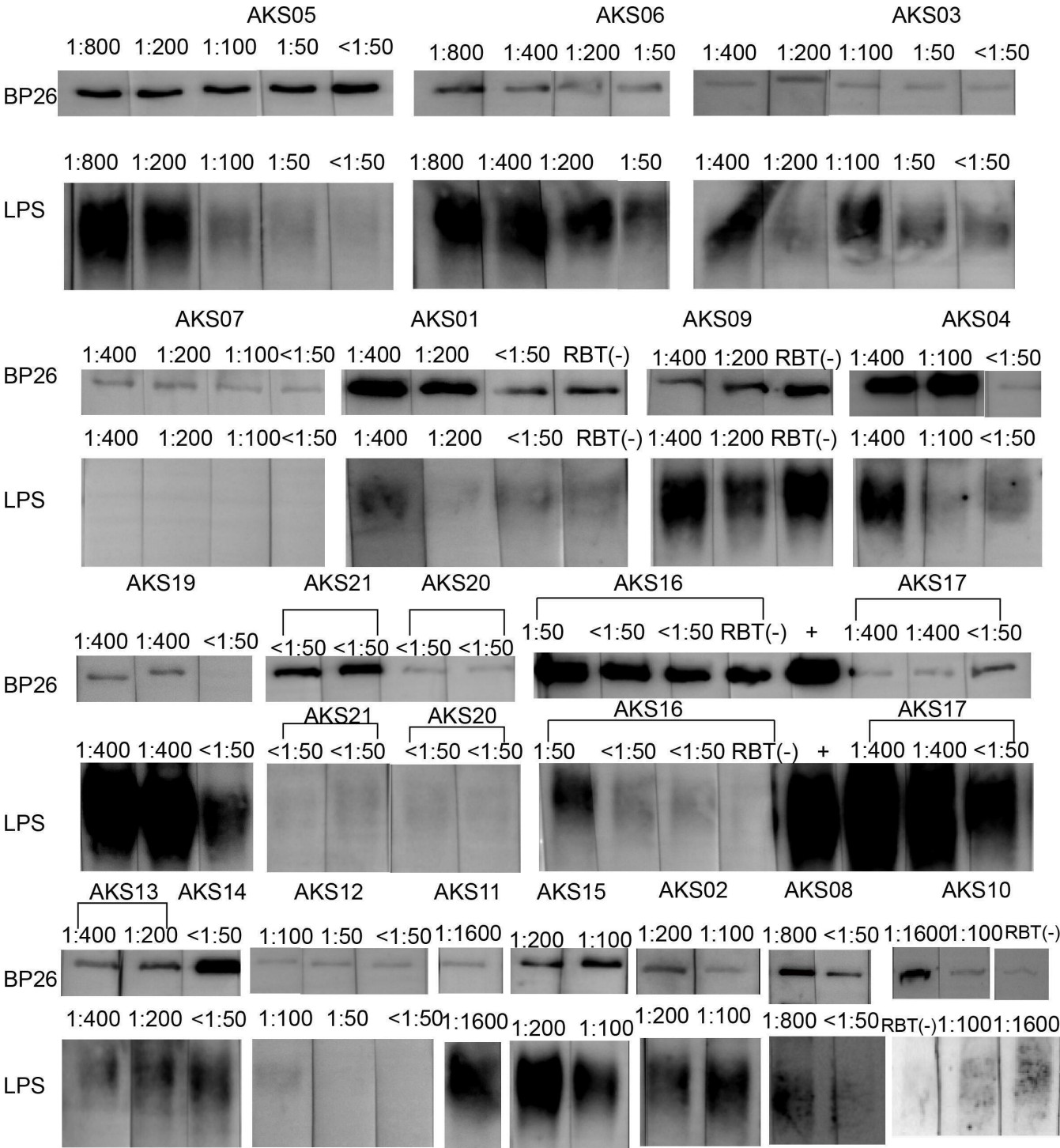

**FIG 4** Western blotting results: hybridization reaction of BP26 and LPS with serum of brucellosis patients (full-length blots are presented in Fig. S1).

Individual immune variability, particularly in immunocompromised individuals or those with genetic polymorphisms affecting antibody production, may impair the humoral response. Additionally, concurrent infections or comorbidities can further modify immune responses, complicating the detection of specific antibodies for brucellosis. Sample integrity is also critical, as improper handling or prolonged storage may compromise antibody stability. Technical factors, such as antigen purity and suboptimal assay conditions, can also impact the reliability of the results. To enhance the precision of

brucellosis detection, we recommend combining western blot results with clinical data and other diagnostic methods. This integrated approach will help mitigate the impact of potential confounders and provide a more comprehensive and accurate diagnosis.

Currently, brucellosis surveillance primarily relies on passive surveillance. Asymptomatic cases are difficult to detect, and untreated patients are at risk of developing chronic complications, which exacerbate the disease burden. Our findings underscore the necessity of strengthening active surveillance, particularly among livestock workers, veterinarians, and meat processing personnel. To improve the accuracy of disease monitoring, we propose a dual-path framework: maintaining existing reporting of symptomatic cases while implementing periodic serological screening for high-risk groups. This integrated approach facilitates comprehensive epidemiological tracking and timely treatment. From a public health perspective, enhanced surveillance should be complemented by community-level education on brucellosis transmission and prevention. However, this study still has some limitations. Single-site design and small sample size limited generalizability. Non-uniform follow-up intervals may introduce variability in antibody kinetics assessment. No pathogen isolation or species-level Brucella identification limited broader applicability. These need to be further improved in future studies.

## Conclusion

This study examined the dynamic progression of brucellosis in Aksai Kazak Autonomous County and provided a scientific foundation for the formulation of targeted prevention and control strategies. Asymptomatic brucellosis patients suggest the need to strengthen *Brucella* infection monitoring in high-risk populations, enabling early detection, diagnosis, and treatment of patients with *Brucella* infection. Analysis of the antibody titers in patients infected with *Brucella* bacteria has revealed that antibodies rise rapidly during the acute phase, with short peak maintenance, followed by a rapid decrease in antibody titers and then a prolonged maintenance time. This means that after treatment, the serum antibody titers of patients in the acute phase can decrease rapidly to lower levels, and patients with acute brucellosis should be treated as early as possible to prevent chronic brucellosis. In addition, these results can provide a basis for future research on the immune response to *Brucella* and the detection of chronic brucellosis.

## MATERIALS AND METHODS

### The data source for an overview of brucellosis in China

Data on human brucellosis cases in China from 2010 to 2024 were obtained from the disease surveillance data of the Chinese Center for Disease Control and Prevention.

### Study subjects

Brucellosis infection among animal husbandry practitioners in Altun Mountain, Akesai Kazakh Autonomous County, Gansu Province, was monitored from 2022 to 2023. A total of 400 animal husbandry practitioners were screened for brucellosis, and 585 serum samples were collected for serological testing. In individuals who were positive for serum antibodies, venous blood samples were collected multiple times throughout the infection after obtaining informed consent from the patients. In addition, the changes in the serum antibody titers (over 7 days) of four brucellosis patients in a hospital in Beijing were assessed.

### Inclusion criteria

Within the study area, participants were required to have direct contact with animals, particularly those involved in farming or herding. Participants who tested positive for

*Brucella* antibodies using both the RBT and SAT were included. Informed consent was obtained from all participants.

## Exclusion criteria

Individuals who tested negative for *Brucella* antibodies in both RBT and SAT and participants who could not provide multiple blood samples over the study period were excluded.

## Follow-up intervals

The initial phase of infection: high-frequency sampling intervals (≤12 days) were prioritized during the initial rapid antibody surge to capture dynamic titer fluctuations.

Convalescent phase monitoring: extended intervals (30 days) were adopted during the gradual antibody decline phase to evaluate long-term immunological trends (12).

## Laboratory detection

### Rose-Bengal test

The RBT was used to screen for *Brucella* antibody. A total of 30 µL of antigen (Idexx Laboratories, Westbrook, Maine, USA) and 30 µL of inactivated serum samples were mixed in a plate, and the results were observed within 4 min.

### Serum agglutination test

The SAT was used to detect the antibody titers of the RBT-positive samples. The serum was serially diluted in turbidimetric tubes (1:50, 1:100, 1:200, 1:400, 1:800, and 1:1,600), and then an equal volume of brucellosis antigen (China Institute of Veterinary Drug Control) was added to each tube. After mixing, the samples were incubated at 37°C for 24 h. By comparing the turbidity of each tube containing a diluted sample and a reference tube, the titers and degree of agglutination (0, +, ++, +++, ++++) were determined. The majority of people in this region rely on animal husbandry for their livelihood and have a clear epidemiological exposure history. In this study, we adopted an SAT titer of ≥1:50++ to classify positive cases in order to more comprehensively identify potential infections.

### Western blotting

Western blotting was performed to detect specific antibodies against BP26 and LPS. The presence of bands at 45.3 kDa (BP26) and 30 kDa (LPS) was recorded. First, 400 ng LPS (purified from *Brucella abortus S19*, purity: ≧95% and concentration: 6.0 mg/mL, stored at 2°C–8°C for long-term storage) and BP26 protein (expressed by *Escherichia coli*, purity: ≧95% and concentration: 1.0 mg/mL, store at 2°C–8°C for not more than 2 weeks, or −20°C or less for long-term storage) (Shanghai Yuduo Biotechnology Co., Ltd., China) were subjected to sodium dodecyl sulfate-polyacrylamide gel electrophoresis and transferred to a polyvinylidene difluoride membrane. The membranes were then incubated with the *Brucella* antibody-positive serum, which was diluted to 1:1,000 with 5% skim milk for 2 h at room temperature, washed with 0.05% Tween in phosphate-buffered saline, and incubated with donkey anti-human IgG secondary antibody (1:10,000 dilution; The Jackson Laboratory) for 1 h. Negative control sera were collected from seronegative healthy adult donors (negative for SAT/RBT) with no history of *Brucella* infection or febrile illnesses, and without known exposure to livestock, raw dairy products, or occupational exposure to rule out nonspecific binding. The protein bands were visualized by adding immunochemiluminescent reagents.

## ACKNOWLEDGMENTS

We thank the Charlesworth Group's author services (paper no. 128782) for their critical editing and helpful comments regarding our manuscript.

This work was supported by the National Key Research and Development Program of China (2022YFC2602203).

Conceptualization: H.S., H.J., and X.W.; Data curation: H.S., Q.D., D.L., F.Q., X.Z., X.L., and A.B.; Formal analysis: H.S., J.G., Z.H., R.D., S.Q., S.G., P.Z., and M.X.; Funding acquisition: X.W. Investigation: H.S., Q.D., D.L., F.Q., X.Z., J.G., Z.H., X.L., A.B., R.D., and S.Q.; Project administration: X.W. Resources: F.Q., X.Z., X.L., and H.J. Supervision: X.W. Validation: Q.D., D.L., F.Q., X.Z., H.J., and X.W; Visualization: H.S.; Writing—original draft: H.S., Q.D., D.L., F.Q., X.Z., and H.J. Writing—review and editing: J.G., Z.H., X.L., A.B., R.D., S.Q., S.G., P.Z., M.X., and X.W. All authors read and approved the final manuscript.

## AUTHOR AFFILIATIONS

[1]National Institute for Communicable Disease Control and Prevention, Chinese Center for Disease Control and Prevention, Beijing, China

[2]Department of Infectious Diseases, Beijing Ditan Hospital, Capital Medical University, Beijing, China

[3]Akesai Kazakh Autonomous County Center for Disease Control and Prevention, Jiuquan, China

[4]Hangzhou Center for Disease Control and Prevention, Hangzhou, China

[5]Tianjin Center for Disease Control and Prevention, Tianjin, China

## AUTHOR ORCIDs

Xin Wang  http://orcid.org/0000-0003-4927-4662

## DATA AVAILABILITY

The datasets used and/or analyzed during the current study are available from the corresponding author upon reasonable request.

## ETHICS APPROVAL

The study was approved by the ethics committee of Akesai Kazakh Autonomous County Center for Disease Control and Prevention, Jiuquan, China (2024-016). Informed consent was obtained from the patients.

## ADDITIONAL FILES

The following material is available online.

### Supplemental Material

**Figure S1 (Spectrum02862-24-S0001.pdf).** Western blot full-length image of Figure 4.
**Supplemental material legends (Spectrum02862-24-S0002.docx).** Legends for Figure S1 and Table S1.
**Table S1 (Spectrum02862-24-S0003.docx).** Demographics table of brucellosis cases in Akesai Kazakh Autonomous County.

### Open Peer Review

**PEER REVIEW HISTORY (review-history.pdf).** An accounting of the reviewer comments and feedback.

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
