## [Reviewer comments · Microbiology Spectrum]

Microbiology Spectrum

Follow-up of antibody changes in brucellosis patients in Gansu, China

Hanyu Sha, Qun Duan, Dongyue Lyu, Fang Qian, Xiaojin Zheng, Jiazhen Guo, Zhaokai He, Xinmin Lu, Asaiti Bukai, shuai qin, Ran Duan, Saier Guli, Peng Zhang, Meng Xiao, Huaiqi Jing, and Xin Wang

Corresponding Author(s): Xin Wang, National Institute for Communicable Disease Control and Prevention

Review Timeline:

Submission Date:	November 8, 2024
Editorial Decision:	January 6, 2025
Revision Received:	January 28, 2025
Editorial Decision:	February 21, 2025
Revision Received:	February 28, 2025
Editorial Decision:	March 4, 2025
Revision Received:	March 16, 2025
Accepted:	March 31, 2025

Editor: Siu-Kei Chow

Reviewer(s): Disclosure of reviewer identity is with reference to reviewer comments included in decision letter(s). The following individuals involved in review of your submission have agreed to reveal their identity: Mustafa Kürşat Şahin (Reviewer #2)

Transaction Report:

DOI: <https://doi.org/10.1128/spectrum.02862-24>

Re: Spectrum02862-24 (**Follow-up of antibody changes in brucellosis patients in Gansu, China**)

Dear Prof. Xin Wang:

Thank you for the privilege of reviewing your work. Below you will find my comments, instructions from the Spectrum editorial office, and the reviewer comments.

Revision Guidelines

Sincerely,
Siu-Kei Chow
Editor
Microbiology Spectrum

Reviewer #1 (Comments for the Author):

Thank you for the opportunity to review the manuscript "Follow-up of antibody changes in brucellosis patients in Gansu, China" which describes the progression of brucellosis by analyzing serum antibody dynamics in patients from Gansu Province, China. Acute brucellosis followed three antibody response phases: rapid rise above 1:200, a decline below this level within 19-41 days, and a slow decline with antibody disappearance over 42-148 days. Western blot analysis revealed the persistence of anti-BP26 protein antibodies, highlighting a durable immune response. These findings may be of interest to the readership as they enhance our current understanding of brucellosis progression. These results may support the monitoring of disease and treatment efficacy, and further analysis of BP26 and LPS may provide a theoretical foundation for vaccine development and

improved therapeutic strategies. The manuscript is well-written and easy to follow. My concerns are outlined below.

Major Concerns:

The small sample size is a major concern, particularly the limited number of antibody-positive individuals (21 out of 400 participants). While the findings on antibody dynamics and immune responses are intriguing, the small cohort may not adequately represent the broader population affected by brucellosis. This could limit the generalizability of the results, as individual variability in immune responses and disease progression might be underrepresented. This should be addressed in the Limitations section of the manuscript, which I found to be lacking. Please include and address any limitations of the study, including differing time intervals of sample collection, location (single county), and culture capabilities to determine if a species component impacted the antibody responses.

Lines 134-138: The inclusion of data from five subjects from a different locale without providing a clear rationale raises concerns about the study's design and the relevance of this subset to the overall findings. It is unclear how these subjects contribute to the study's objectives or their inclusion's impact on the interpretation of results. This lack of explanation makes it difficult to assess whether the additional data enhances the study's robustness or merely introduces variability. Justifying their inclusion, such as unique epidemiological factors or differences in exposure patterns, would help clarify their role and ensure that their data aligns meaningfully with the study's aims.

Minor Concerns:

Line 38: "lasted for a long time". Please quantify and provide a value as a mean or median

Line 92: Suggest referencing the publication by Wang, 2022, Emerging Infectious Diseases

Line 96: Consider removing panels 1B/1C, I am not sure they add to the results.

Lines 101-111: Was the exposure for all participants direct animal exposure? Could this be added to Table 1 or to a demographics table as suggested below?

Line: 115: Is (1) meant for a reference or something else? Same for (2) on line 119.

Line 132-133: Please provide more information regarding the chronic participants (AKS20 and AKS21) as little information was provided and the results are briefly addressed in the discussion. Please include why they are not represented in Figure 2.

Line 162-164: If the exposure type was different for the asymptomatic cases could this have had any impact on the lower antibody titers? Please address.

Line 185-186: Do the authors have any explanation for why two patients were negative for anti-LPS or what the implications are for use as a clinical diagnostic tool?

Table 1: Consider moving to supplemental and replacing it with a demographics table which includes the ages of the participants.

Fig 2: Panels AKS02 and AKS17 have wording on the y-axis that is not in the other panels. Time intervals in red seem to be missing from AKS03.

Fig 4: Please address the missing blots for AKS08, AKS10, AKS18.

Reviewer #2 (Comments for the Author):

Study Relevance and Context:

1. The manuscript effectively emphasizes the significance of brucellosis as a public health concern. However, consider elaborating on why Gansu, China, was selected for this study, highlighting specific epidemiological challenges or gaps in existing research that your study addresses.

Methodological Details:

1. Clarify why certain patients' antibody titers were not analyzed (e.g., AKS18, AKS20, and AKS21 lack some results in Table 2). Were these omissions due to logistical constraints, patient condition, or another reason?

2. Expand on the rationale behind using specific thresholds (e.g., {greater than or equal to}1:50++ in SAT) to classify positive cases.

3. Please provide supplementary methodological details (e.g., detailed inclusion criteria, antibody variability metrics) to strengthen reproducibility.

4. Statistical analyses are not fully discussed, particularly regarding the generalizability of findings from a small cohort (21 positive cases out of 400). Addressing the impact of sampling on statistical power is recommended.

Data Presentation:

1. Tables and figures are informative but could benefit from additional labeling. For example, Figure 1 should explicitly state whether trends are cumulative or annual. Similarly, Figures 3 and 4 would benefit from annotations linking bands to specific findings in the text.

Results Interpretation:

1. Discuss the implications of asymptomatic cases in greater depth. What do these findings suggest about potential undiagnosed reservoirs in other high-risk groups?
2. Please include greater integration of local findings with global epidemiological frameworks to enhance the article's relevance to an international audience.
3. The manuscript mentions cross-reactivity in serological tests. Highlight whether and how these limitations affected your conclusions or necessitated follow-up confirmatory tests.

Conclusion and Implications:

1. While the discussion addresses the clinical and epidemiological relevance of findings, consider tying these insights to global public health strategies. For instance, how could these results influence international vaccination efforts or diagnostic improvements?
2. Statistical analyses are not fully discussed, particularly regarding the generalizability of findings from a small cohort (21 positive cases out of 400). Addressing the impact of sampling on statistical power is recommended.
3. While the Western blot analysis adds depth, the lack of *Brucella* species identification limits the broader applicability of the findings. Please address.

Reviewer #1 (Comments for the Author):

Major Concerns:

Question 1. The small sample size is a major concern, particularly the limited number of antibody-positive individuals (21 out of 400 participants). While the findings on antibody dynamics and immune responses are intriguing, the small cohort may not adequately represent the broader population affected by brucellosis. This could limit the generalizability of the results, as individual variability in immune responses and disease progression might be underrepresented. This should be addressed in the Limitations section of the manuscript, which I found to be lacking. Please include and address any limitations of the study, including differing time intervals of sample collection, location (single county), and culture capabilities to determine if a species component impacted the antibody responses.

Answer 1. We acknowledge the limitations of our study, particularly the small sample size, variability in sample collection timing, geographical constraints, and challenges in culturing and identifying *Brucella* species. These limitations may affect the generalizability of our findings and the interpretation of antibody dynamics. Despite these limitations, our study offers valuable insights into the progression of brucellosis from an immunological perspective by analyzing the dynamic changes in serum antibodies of patients. This analysis may support disease monitoring and the evaluation of treatment efficacy. We have included a detailed discussion of these limitations in the manuscript, specifically in lines 266-281, to provide a transparent account of our study's scope and potential areas for improvement.

Question 2. Lines 134-138: The inclusion of data from five subjects from a different locale without providing a clear rationale raises concerns about the study's design and the relevance of this subset to the overall findings. It is unclear how these subjects contribute to the study's objectives or their inclusion's impact on the interpretation of results. This lack of explanation makes it difficult to assess whether the additional data enhances the study's robustness or merely introduces variability. Justifying their inclusion, such as unique epidemiological factors or differences in exposure patterns, would help clarify their role and ensure that their data aligns meaningfully with the study's aims.

Answer 2. We sincerely appreciate your valuable comments. We acknowledge that the five subjects from a different locale in our study are not directly comparable to those within the study area. It should also be clarified that these five subjects were not included in our follow-up cohort, and the data they provided were used solely for reference purposes. Specifically, the inclusion of these five subjects was primarily based on the following two key factors:

1. We successfully isolated the *Brucella* pathogen from one of the participants. This finding serves as a valuable reference for diagnosing brucellosis in our study, which we have detailed on lines 228-231.

2. During the seven-day observation period, the antibody titers of the five participants from outside the study area remained remarkably stable, showing almost no significant changes. This important finding offers valuable insights for determining the follow-up intervals for *Brucella* patients within the study area, as noted on lines 238-242.

Of course, we are also keenly aware of the limitations of our study. Selecting patients directly from within the study area and conducting short-term follow-up observations may yield more

effective and convincing results. Additionally, a more uniform and intensive sampling strategy could provide more accurate and reliable outcomes.

Minor Concerns:

Question 1. Line 38: "lasted for a long time". Please quantify and provide a value as a mean or median

Answer 1. Thanks for your suggestion, we have also quantified this result to "lasted for a long time(up to 395 days)", on line 38.

Question 2. Line 92: Suggest referencing the publication by Wang, 2022, Emerging Infectious Diseases

Answer 2. Thanks for your comments, we have added the reference by Wang, 2022, Emerging Infectious diseases on line 103 of the manuscript.

Question 3. Line 96: Consider removing panels 1B/1C, I am not sure they add to the results.

Answer 3. Thank you for your suggestion, we agree and have removed 1B/1C from Figure 1. In addition, according to the latest national notifiable infectious disease epidemic situation released by the Monitoring and Early Warning Department of the National Disease Prevention and Control Bureau, the number of brucellosis cases in China for 2024 has been added in Figure 1.

Question 4. Lines 101-111: Was the exposure for all participants direct animal exposure? Could this be added to Table 1 or to a demographics table as suggested below?

Answer 4. Thank you for your valuable comments. All participants included in this study had direct contact with animals. We have also revised the original Table 1 to a supplementary table in response to question 9, which is a demographic table indicating whether each participant has direct contact with animals (see lines 113-116).

Question 5. Line: 115: Is (1) meant for a reference or something else? Same for (2) on line 119.

Answer 5. Thank you for your comments, and we are very sorry that the number we used in the manuscript is confused with the reference label, which has been replaced by ①, ②, ③ in the manuscript lines 127-133.

Question 6. Line 132-133: Please provide more information regarding the chronic participants (AKS20 and AKS21) as little information was provided and the results are briefly addressed in the discussion. Please include why they are not represented in Figure 2.

Answer 6. Thank you for your valuable advice. We have incorporated clinical symptom information for chronic participants (AKS20 and AKS21) into the results section of the manuscript, specifically on lines 145-147. Additionally, we have expanded the discussion of the results on lines 254-260 to provide a more comprehensive analysis. Furthermore, we have included a line chart for the chronic participants (AKS20 and AKS21) in Figure 2, illustrating the changes in serum antibody titers throughout the follow-up period.

Question 7. Line 162-164: If the exposure type was different for the asymptomatic cases could this have had any impact on the lower antibody titers? Please address.

Answer 7. Thank you for your valuable comments. Brucellosis can be transmitted through multiple routes, including skin contact, ingestion via the digestive tract, and inhalation through the respiratory tract. Different routes of infection may result in varying distributions and degrees of *Brucella* infection within the body, which can subsequently influence the production of antibody titers. One study found that different types of exposure—such as contact with pathogens through various pathways or at different doses—lead to distinct antibody responses[1]. For instance, an individual who inhales the pathogen through the respiratory tract may produce a higher antibody titer, whereas an individual who ingests it through the digestive tract may generate a lower antibody titer. In addition, the size of the infectious dose also affects antibody production. Generally, the larger the infectious dose, the higher the antibody titer is likely to be. Factors such as age, sex, and the immune status of the infected individual can also influence antibody production. For example, younger individuals and those with stronger immune systems may produce higher antibody titers. Therefore, we have added the relevant discussion the manuscript lines 192-194.

References

[1]Uslu, A., Sayın, Z., Balevi, A., Gulcu, Y., Ergen, F., Akıner, I., Denizli, O., & Erganis, O. (2025). Comparison of Long-Term Antibody Titers in Calves Treated with Different Conjunctival and Subcutaneous *Brucella abortus* S19 Vaccines. *Animals : an open access journal from MDPI*, 15(2), 212. <https://doi.org/10.3390/ani15020212>

Question 8. Line 185-186: Do the authors have any explanation for why two patients were negative for anti-LPS or what the implications are for use as a clinical diagnostic tool?

Answer 8. Thanks to your careful review and advice, we explain the negative anti-LPS results in two patients (AKS07 and AKS12).

①Low antibody titers:

Both patients (AKS07 and AKS12) had low antibody titers, which may have contributed to the negative results in the anti-LPS Western blot analysis. Low titers can indicate a weaker immune response, potentially leading to undetectable levels of anti-LPS antibodies.

②Variability in immune response:

The immune response to *Brucella* infection can vary significantly among individuals. Some patients may produce higher levels of antibodies against specific antigens like BP26, while others may have a more muted response to LPS. This variability can be influenced by factors such as the infecting strain, the route of infection, and the individual's immune status.

③Timing of samples collection:

The timing of blood sample collection relative to the onset of infection can affect antibody detection. If samples were collected early in the infection, antibodies against LPS might not have reached detectable levels. Conversely, if the samples were collected late in the infection, antibodies might have waned.

④Preservation conditions of serum samples :

The preservation conditions of serum samples are crucial for the maintenance of antibody activity. If the sample is stored at room temperature for too long or the storage temperature fluctuates greatly, the antibody degradation or inactivation may occur. In addition, repeated freezing and thawing will also reduce the titer of antibodies in the serum, thereby increasing the risk of false negative antibody tests.

Combining anti-LPS testing with other antigens, such as BP26, can enhance diagnostic accuracy [1]. In our study, BP26 antibodies were detected in all patients, suggesting that BP26 may serve as a more reliable marker for the diagnosis of brucellosis. We have already included possible explanations for the negative anti-LPS results of AKS07 and AKS12 in the discussion section, lines 218-224.

Reference

[1]Xie, Y., Lin, S., Guo, L., Qi, X., Zhao, S., Pei, Q., Chen, Y., Wu, Q., Wang, Y., Yao, M., & Yin, D. (2025). Development and evaluation of the recombinant BP26 protein-based C-ELISA for human brucellosis diagnosis. *Frontiers in microbiology*, 15, 1516915. <https://doi.org/10.3389/fmicb.2024.1516915>

Question 9. Table 1: Consider moving to supplemental and replacing it with a demographics table which includes the ages of the participants.

Answer 9. Thank you for your valuable comments. And we have changed Table 1 to supplemental and replacing it with a demographics table which includes the ages of the participants, lines 113-114.

Question 10. Fig 2: Panels AKS02 and AKS17 have wording on the y-axis that is not in the other panels. Time intervals in red seem to be missing from AKS03.

Answer 10. Thank you for your suggestion. We have removed the words of Y-axis of AKS02 and AKS17 in Figure 2 and added the time interval of AKS03. In addition, we have added the line chart of antibody titer changes of AKS18, AKS19, AKS20 and AKS21 that were previously missing.

Question 11. Fig 4: Please address the missing blots for AKS08, AKS10, AKS18.

Answer 11. Thank you very much for your valuable suggestions. We have carefully reorganized the original images from the Western blot experiment. For the previously missing blots of AKS08 and AKS10, we have now supplemented and improved them in Figure 4. However, we regret that due to our inability to obtain the serum sample of AKS18 during the experiment, we cannot provide the corresponding missing blot. Additionally, we were unable to successfully acquire serum samples of certain titers for some patients, which means we cannot provide the corresponding blot images. We have clearly explained this situation in the results section of the manuscript, specifically in lines 164-167, to ensure that readers fully understand the limitations of the experimental results and the integrity of the data.

We are aware that the absence of certain data may affect the comprehensiveness and accuracy of the experiment. Nevertheless, we have made every effort to present the most reliable and precise data given the current circumstances. We look forward to addressing these limitations in future research and offering more complete and in-depth insights for the relevant fields. Thank you once again for your understanding and support.

Reviewer #2 (Comments for the Author):

Study Relevance and Context:

Question 1. The manuscript effectively emphasizes the significance of brucellosis as a public health concern. However, consider elaborating on why Gansu, China, was selected for this study, highlighting specific epidemiological challenges or gaps in existing research that your study addresses.

Answer 1. Thank you for your valuable comments. We have provided a detailed explanation in the introduction section of the manuscript, specifically in lines 73-86, regarding the selection of Gansu, China, as the study area for this research. Firstly, the prevalence of brucellosis in Gansu Province cannot be overlooked. Located in northwest China, Gansu is characterized by a diverse ethnic population and a relatively underdeveloped economy. Due to its complex geographical environment, well-developed animal husbandry, and relatively scarce medical and health resources, the incidence of brucellosis in this area remains high[1]. Secondly, Gansu Province faces specific epidemiological challenges, particularly in the focus of this study, the Kazakh Autonomous County. The residents here primarily depend on animal husbandry for their livelihoods, creating a high-risk population and a concentrated area of high-risk behaviors. However, it is concerning that the local population's awareness of brucellosis is still inadequate. Additionally, the vast territory, sparse population, and varied terrain, combined with the frequent migration of herdsmen between pastures, pose significant difficulties and challenges for the implementation of disease control measures. Therefore, conducting in-depth research on the prevalence of brucellosis in this region of Gansu is crucial for formulating effective control strategies and reducing the disease burden. This research can also provide valuable references and insights for other regions, collectively advancing the progress and development of brucellosis control efforts.

Reference

[1]Wang Pinggui, He Aiwei, Niu Lixia, Zhou Xiaoyan, Zhao Qi, Ma Minghui, Guan Hong, Feng Yu. Investigation of capability and performance of brucellosis detection in medical and health institutions in Gansu, 2018–2022[J]. Disease Surveillance, 2024, 39(11): 1429-1433. DOI: 10.3784/jbjc.202403010134 [in Chinese]

Methodological Details:

Question 1. Clarify why certain patients' antibody titers were not analyzed (e.g., AKS18, AKS20, and AKS21 lack some results in Table 2). Were these omissions due to logistical constraints, patient condition, or another reason?

Answer 1. Thank you very much for your valuable suggestions. The absence of Western blot results in Table 2(Now it becomes Table1) is due to logistical constraints. Specifically, we faced challenges in obtaining serum samples with certain titers from some patients. As a result, we were unable to conduct the necessary analyses for these specific titers. We have made every effort to minimize the impact of these omissions on the study outcomes. Furthermore, we have clearly explained this situation in the results section of the manuscript (lines 164-167) to ensure that readers fully understand the limitations of the experimental results and the integrity of the data.

Question 2. Expand on the rationale behind using specific thresholds (e.g., {greater than or equal to} 1:50++ in SAT) to classify positive cases.

Answer 2. Thank you for your valuable suggestions. According to the Chinese National Health

Industry Standard (WS 269—2019) titled "Diagnosis of Brucellosis," the criteria for laboratory confirmation of Brucella infection are as follows:

For acute-phase infections, an SAT titer of 1:100++ or higher.

For infections lasting more than one year with persistent clinical symptoms, an SAT titer of 1:50++ or higher.

Given that the majority of individuals in this region depend on animal husbandry for their livelihoods and possess a well-defined epidemiological exposure history, this study adopted an SAT titer of $\geq 1:50++$ to classify positive cases. This approach aims to more comprehensively identify potential infections. This criterion not only adheres to the national health industry standards but also considers the specific epidemiological context of the region. It enhances both the sensitivity and specificity of diagnoses, ensuring the timely detection and management of potential infection sources (lines 326-329).

Question 3. Please provide supplementary methodological details (e.g., detailed inclusion criteria, antibody variability metrics) to strengthen reproducibility.

Answer 3. Thank you for your valuable suggestions. To enhance the reproducibility of our study, we have provided additional supplementary methodological details in the manuscript on lines 307-314 and 326-332.

Inclusion Criteria:

Occupational Exposure: Participants were required to have direct contact with animals, particularly those involved in agriculture or livestock herding.

Serological Evidence: Participants who tested positive for Brucella antibodies using both the Rose–Bengal test (RBT) and the serum agglutination test (SAT) were included in the study.

Informed Consent: Informed consent was obtained from all participants.

Exclusion Criteria:

Lack of Serological Evidence: Individuals who tested negative for Brucella antibodies in both RBT and SAT were excluded due to a lack of serological evidence.

Inability to Provide Follow-Up Samples: Participants who were unable to provide multiple blood samples throughout the study period were excluded.

Western Blot:

Western blotting was conducted to detect specific antibodies against BP26 and LPS. The presence of bands at 45.3 kDa (BP26) and 30 kDa (LPS) was observed.

Serum Agglutination Test (SAT):

We adopted an SAT titer of $\geq 1:50++$ to classify positive cases, aiming to more comprehensively identify potential infections. This criterion not only adheres to the national health industry standard but also considers the specific epidemiological context of the region.

Question 4. Statistical analyses are not fully discussed, particularly regarding the generalizability of findings from a small cohort (21 positive cases out of 400). Addressing the impact of sampling on statistical power is recommended.

Answer 4. We sincerely appreciate your thorough review of our study and acknowledge your concern regarding the insufficient discussion of statistical analysis and the potential impact of small cohort sample sizes on the generalizability of our results. In our study, we based our analysis on a relatively small cohort, consisting of only 21 positive cases out of 400 subjects, which may

have somewhat limited the generalizability of our findings. However, it is important to note that this study area possesses unique characteristics. The region is vast and sparsely populated, resulting in a relatively low population density. Although the sample size is limited, our study benefits from an extended follow-up period, with the longest follow-up reaching 489 days. This duration provides valuable data to support a comprehensive investigation of disease progression and treatment effects. We have elaborated on this in the discussion section of the manuscript, specifically in lines 269-274, to provide readers with a comprehensive understanding of this potential limitation.

Data Presentation:

Question 1. Tables and figures are informative but could benefit from additional labeling. For example, Figure 1 should explicitly state whether trends are cumulative or annual. Similarly, Figures 3 and 4 would benefit from annotations linking bands to specific findings in the text.

Answer 1. We appreciate the suggestion to enhance the informativeness of our figures. In response to these concerns, we have made the following revisions: We have added a clear label to specify that the trends shown in Figure 1 represent the annual number of brucellosis cases in China from 2010 to 2024. This clarification will help readers understand whether the data reflects cumulative totals or annual incidence rates. To improve the connection between the Western blot results and the specific findings discussed in the text, we have added annotations to the figures. These annotations link the observed bands to the corresponding antibody titers and specific patient samples (e.g., AKS01, AKS02, etc.). In Figure 3, we have included labels to indicate the specific molecular weights of the target bands for BP26 (45.3 kDa) and LPS (30 kDa). In Figure 4, we have annotated the blots to highlight the presence or absence of bands for each patient sample, correlating these results with the antibody titers described in the text.

Results Interpretation:

Question 1. Discuss the implications of asymptomatic cases in greater depth. What do these findings suggest about potential undiagnosed reservoirs in other high-risk groups?

Answer 1. Thank you very much for your valuable comments and suggestions. We have discussed asymptomatic cases in greater depth and detail in the discussion section, specifically in lines 183-188 and 190-194. Based on the original content, we have expanded and elaborated on the following dimensions:

①Hidden Disease Burden: Asymptomatic cases, which accounted for 28.57% (6 out of 21) of our positive cases, suggest that brucellosis may be significantly underdiagnosed. These individuals, despite being infected with *Brucella*, did not exhibit clinical symptoms at the time of testing. This indicates that the true prevalence of brucellosis might be higher than currently reported, as asymptomatic carriers are less likely to seek medical attention and be tested.

②Broader Screening Needed: Our findings suggest that similar undiagnosed reservoirs may exist in other high-risk populations. For instance, in regions with intensive animal husbandry, individuals who handle livestock products (e.g., milk, meat) or work in slaughterhouses may be at high risk of infection but remain undiagnosed due to the lack of symptoms. This highlights the importance of expanding screening efforts beyond symptomatic individuals to include those with significant occupational exposure.

Question 2. Please include greater integration of local findings with global epidemiological frameworks to enhance the article's relevance to an international audience.

Answer 2. Thank you for your valuable comments. To enhance the relevance of the article to an international audience, the discussion section of the manuscript (lines 169-172 and lines 177-182) provides a brief overview of the global prevalence of brucellosis, underscoring its significance as a major zoonotic disease that poses a threat to public health. The local results were compared with the global context, revealing that the incidence rate in Aksai Kazakh Autonomous County, Gansu Province (5.25%), is comparable to that of other high-incidence regions worldwide, such as Mongolia, the Mediterranean region, and the Middle East. This comparison emphasizes the importance of this region in the global brucellosis epidemic.

Question 3. The manuscript mentions cross-reactivity in serological tests. Highlight whether and how these limitations affected your conclusions or necessitated follow-up confirmatory tests.

Answer 3. Thank you for your valuable comments. In this study, the BP26 antigen was utilized for Western blot analysis, which effectively minimizes cross-reactivity in serological testing for brucellosis, thereby enhancing the accuracy and reliability of diagnosis. BP26 is a major immunogenic protein of *Brucella* and demonstrates excellent species specificity. Compared to the traditional Serum Agglutination Test (SAT), BP26 as a diagnostic antigen significantly decreases cross-reactivity with other pathogens, such as *Yersinia enterocolitica* [1]. This specificity makes BP26 more accurate in detecting *Brucella* infections. Compared to other outer membrane proteins of *Brucella*, such as OMP31, BP26 exhibits higher diagnostic specificity. In a comparative study, the diagnostic accuracy rates of BP26 for human and goat brucellosis sera were 96.45% and 95.00%, respectively, while the diagnostic accuracy rate of OMP31 for bovine serum was 84.03%[2]. Additionally, BP26 demonstrated a lower false-positive rate in cross-reactivity experiments using sera from non-*Brucella* infections.

Reference

[1]Tian, M., Song, M., Yin, Y., Lian, Z., Li, Z., Hu, H., Guan, X., Cai, Y., Ding, C., Wang, S., Li, T., Qi, J., & Yu, S. (2020). Characterization of the main immunogenic proteins in *Brucella* infection for their application in diagnosis of brucellosis. *Comparative immunology, microbiology and infectious diseases*, 70, 101462. <https://doi.org/10.1016/j.cimid.2020.101462>

[2]Bai, Q., Li, H., Wu, X., Shao, J., Sun, M., & Yin, D. (2021). Comparative analysis of the main outer membrane proteins of *Brucella* in the diagnosis of brucellosis. *Biochemical and biophysical research communications*, 560, 126–131. <https://doi.org/10.1016/j.bbrc.2021.04.127>

Conclusion and Implications:

Question 1. While the discussion addresses the clinical and epidemiological relevance of findings, consider tying these insights to global public health strategies. For instance, how could these results influence international vaccination efforts or diagnostic improvements?

Answer 1. Thank you for your valuable advice. We have connected these insights to the global public health strategy in the discussion section, specifically in lines 211-224, in two ways:

①Vaccine Development and International Vaccination Efforts:

The study highlights the persistence of antibodies against the BP26 protein in brucellosis patients,

suggesting its potential as a vaccine candidate antigen. As a major immunogenic protein of *Brucella*, BP26 could induce long-lasting protective immunity. This insight may guide the development of vaccines that are effective against *Brucella* infections, particularly in high-risk populations such as farmers, herders, and veterinarians.

②Diagnostic Improvements:

Enhanced Specificity with BP26: The study demonstrates that BP26 can be utilized in Western blot analysis to minimize cross-reactivity with other pathogens, thereby improving the specificity of serological tests. Additionally, combining anti-LPS testing with BP26 can enhance diagnostic accuracy.

Question 2. Statistical analyses are not fully discussed, particularly regarding the generalizability of findings from a small cohort (21 positive cases out of 400). Addressing the impact of sampling on statistical power is recommended.

Answer 2. We sincerely appreciate your thorough review of our study and acknowledge your concern regarding the insufficient discussion of statistical analysis and the potential impact of small cohort sample sizes on the generalizability of our results. In our study, we based our analysis on a relatively small cohort, consisting of only 21 positive cases out of 400 subjects, which may have somewhat limited the generalizability of our findings. However, it is important to note that this study area possesses unique characteristics. The region is vast and sparsely populated, resulting in a relatively low population density. Although the sample size is limited, our study benefits from an extended follow-up period, with the longest follow-up reaching 489 days. This duration provides valuable data to support a comprehensive investigation of disease progression and treatment effects. We have elaborated on this in the discussion section of the manuscript, specifically in lines 269-274, to provide readers with a comprehensive understanding of this potential limitation.

Question 3. While the Western blot analysis adds depth, the lack of *Brucella* species identification limits the broader applicability of the findings. Please address.

Answer 3. Thank you for your valuable feedback. The biological characteristics of *Brucella*, including its slow growth rate in vitro, strict nutritional requirements, and susceptibility to various factors such as the stage of infection, site of infection, pathogen load in samples, and prior antibiotic use, make it particularly challenging to isolate *Brucella* pathogens directly from human samples. Although we employed a multiplex PCR method to detect nucleic acids in the blood samples of participants, this method also failed to identify the specific species of *Brucella*. Therefore, we acknowledge that the absence of *Brucella* species identification in our study limits the broader applicability of our findings. We have already illustrated this limitation in the discussion section of our manuscript, specifically in lines 266-281. Despite this limitation, our study provides valuable insights into the progression of brucellosis from an immunological perspective by analyzing the dynamic changes in serum antibodies of patients. The Western blot analysis showed the persistence of antibodies against the BP26 protein, which may support disease monitoring and the evaluation of treatment efficacy.

Re: Spectrum02862-24R1 (**Follow-up of antibody changes in brucellosis patients in Gansu, China**)

Dear Prof. Xin Wang:

Thank you for the privilege of reviewing your work. Below you will find my comments, instructions from the Spectrum editorial office, and the reviewer comments.

Revision Guidelines

Sincerely,
Siu-Kei Chow
Editor
Microbiology Spectrum

Reviewer #2 (Comments for the Author):

Comments:

Clarify the rationale behind follow-up intervals: The study discusses antibody kinetics but does not clearly justify the chosen time points for follow-up. A brief explanation of why these intervals were selected would improve transparency.

Expand on the diagnostic value of BP26 and LPS: The Western blot results indicate prolonged persistence of BP26 antibodies, which could have diagnostic implications. Discussing how BP26 testing might complement traditional serological methods would enhance the manuscript's impact.

Refine the discussion on asymptomatic cases: The finding that 28.57% of cases were asymptomatic is important for public health. Expanding on how this influences brucellosis screening strategies and disease surveillance would be valuable.

Consider minor English language editing: Some sentences, particularly in the introduction and discussion sections, could be rewritten for better clarity and readability.

Reviewer #2 (Comments for the Author):

Comments:

Question 1: Clarify the rationale behind follow-up intervals: The study discusses antibody kinetics but does not clearly justify the chosen time points for follow-up. A brief explanation of why these intervals were selected would improve transparency.

Answer 1:

The Brucellosis Tube Agglutination Test (SAT) serves as a critical serological assay for detecting pan-immunoglobulin antibodies (IgG, IgM, and IgA) in brucellosis patients. IgM antibodies typically peak within 1-2 weeks post-infection, representing an early diagnostic marker. However, their diagnostic utility diminishes over time due to low antigen-binding affinity and rapid decay kinetics. In contrast, IgG antibodies emerge as the predominant immunoglobulins by weeks 2-3 post-infection, characterized by extended half-life (approximately 21 days), high-affinity binding, and sustained pathogen-neutralizing capacity. Consequently, SAT titers predominantly reflect IgG dynamics, particularly in chronic infections and post-therapeutic monitoring.

Based on these antibody kinetics, our study implemented a dual-phase follow-up protocol (added to lines 338-342 of the manuscript):

Initial phase of infection: High-frequency sampling intervals (≤ 12 days) were prioritized during the initial rapid antibody surge to capture dynamic titer fluctuations.

Convalescent phase monitoring: Extended intervals (30 days) were adopted during the gradual antibody decline phase to evaluate long-term immunological trends.

Notably, logistical constraints inherent to our study region—including vast geographical expanses, low population density, nomadic pastoralist communities, and limited telecommunications infrastructure—complicated strict adherence to sampling schedules. These operational challenges are addressed in the Discussion section of the manuscript, lines 292-300. To validate our interval selection, we referenced antibody titer observations from four brucellosis patients monitored at Beijing Ditan Hospital over 7-day intervals. Notably, three patients exhibited stable titers, while one demonstrated moderate decline (1:800++ to 1:400++), supporting the rationale for extended intervals during non-acute phases.

Future investigations would benefit from systematically designed dense sampling protocols to further optimize monitoring intervals. Such refinements could enhance the precision of brucellosis surveillance and therapeutic efficacy assessments, particularly in resource-limited endemic regions.

Question 2: Expand on the diagnostic value of BP26 and LPS: The Western blot results indicate prolonged persistence of BP26 antibodies, which could have diagnostic implications. Discussing how BP26 testing might complement traditional serological methods would enhance the manuscript's impact.

Answer 2: Thank you for your insightful comments. We have thoroughly expanded on the diagnostic value of BP26 and LPS in the discussion section of the manuscript. Specifically, we have incorporated a detailed discussion of BP26 detection (lines

226-230) to highlight how it can complement traditional serological methods for diagnosing brucellosis in two critical scenarios: (1) when SAT results are negative despite strong clinical suspicion of brucellosis, and (2) during chronic infection monitoring. The extended detection window of BP26 antibodies, attributable to their slower decay kinetics relative to LPS antibodies, enables reliable identification of infections even at antibody titers below the detection thresholds of conventional serological assays.

Question 3: Refine the discussion on asymptomatic cases: The finding that 28.57% of cases were asymptomatic is important for public health. Expanding on how this influences brucellosis screening strategies and disease surveillance would be valuable.

Answer 3: Thanks for your valuable comments, we have refined the discussion of asymptomatic cases in lines 186-194 of the discussion section of the manuscript.

Brucellosis Screening Strategies :

The study found that among serum *Brucella* antibody - positive cases, a significant proportion (28.57%) were asymptomatic individuals detected through active surveillance of high - risk populations. This highlights the importance of active screening. Untreated asymptomatic brucellosis can progress to chronic complications. To improve screening, a dual - path surveillance framework is proposed. It includes maintaining existing symptomatic case reporting systems and implementing mandatory periodic serological screening for high - exposure occupational groups such as livestock workers, veterinarians, and meat processing personnel. This will help in early detection and therapeutic interventions.

Disease Surveillance :

The current passive surveillance systems tend to miss asymptomatic carriers of brucellosis, leading to an underestimation of the true disease prevalence. The proposed dual - path surveillance framework aims to enhance disease monitoring accuracy. By combining the existing symptomatic case reporting with mandatory periodic serological screening for high - risk occupational groups, more comprehensive epidemiological tracking can be achieved. From a public health perspective, this enhanced surveillance should be accompanied by community - level education programs about brucellosis transmission pathways and preventive measures.

Question 4: Consider minor English language editing: Some sentences, particularly in the introduction and discussion

Answer 4: We are very grateful to the comments on this manuscript. As for the language and grammar suggestions, we have reworded the introduction and discussion to improve the readability and accuracy of the manuscript.

Re: Spectrum02862-24R2 (**Follow-up of antibody changes in brucellosis patients in Gansu, China**)

Dear Prof. Xin Wang:

Thank you for the privilege of reviewing your work. Below you will find my comments, instructions from the Spectrum editorial office, and the reviewer comments.

Revision Guidelines

Sincerely,
Siu-Kei Chow
Editor
Microbiology Spectrum

Reviewer #2 (Comments for the Author):

The revised manuscript presents an important study on the dynamics of antibody changes in brucellosis patients in Gansu, China. The authors have made significant efforts to improve clarity, enhance methodological details, and strengthen data interpretation based on the revisions. The study offers valuable insights into serological monitoring and potential diagnostic markers, particularly BP26, for tracking brucellosis progression. The revised version demonstrates a marked improvement in organization, scientific rigor, and clarity. However, some aspects still require refinement for final acceptance.

Revisions Required

Abstract: Improve Clarity and Structure

The abstract is too detailed and includes unnecessary methodological specifics. It should be concise and structured, highlighting the background, objectives, key findings, and conclusions.

Suggested improvements:

Clearly state the importance of monitoring antibody dynamics in brucellosis.

Summarize results more effectively, focusing on key findings without excessive details on methodology.

The last sentence should emphasize the clinical and public health significance rather than restating findings.

Introduction: Enhance Justification and Study Rationale

The introduction provides good background on brucellosis but could be more focused on the study rationale.

The public health importance of understanding antibody kinetics should be better integrated into the objectives.

The section on study site characteristics (Akesai Kazakh Autonomous County) is too detailed. It should be more concisely linked to the study's significance.

Methods: Improve Reproducibility

The sample selection criteria need more clarity. Why was 400 selected as the initial sample size? Provide justification based on statistical or epidemiological considerations.

Western blotting methodology: Provide details on quality control measures for protein purity and specificity.

Follow-up intervals: The rationale for 30-day follow-ups should be justified by referencing relevant literature.

Results: Improve Data Presentation and Interpretation

The section effectively describes antibody kinetics but needs clearer statistical validation.

Figures and tables: Figure legends should be more descriptive.

Table 1 (Western blot results) lacks statistical validation-consider adding statistical tests to compare antibody persistence trends.

The Western blot findings should discuss potential confounders, such as patient immune variability or testing conditions.

Discussion: Strengthen Interpretation and Clinical Implications

The discussion is comprehensive but somewhat redundant. It should be more structured, focusing on:

Key findings and their implications for brucellosis diagnosis and treatment.

Comparison with previous literature-how do these findings align with or contradict other serological studies?

Potential applications of BP26 protein detection-can it be used for early diagnosis or vaccine development?

The implications for brucellosis surveillance and control strategies should be elaborated.

Limitations: Acknowledge Study Constraints More Explicitly

While limitations are mentioned, the following should be explicitly acknowledged:

Small sample size (n=21 positive cases), limiting generalizability.

Non-uniform follow-up intervals, which may introduce variability in antibody kinetics assessment.

No pathogen isolation or species-level *Brucella* identification, which limits broader applicability.

Grammar and Style Improvements

There are grammatical errors and awkward phrasing that need refinement.

Standardized terminology should be consistently used (e.g., "Brucella seropositivity" instead of "Brucella antibody-positive cases").

References

Some references are outdated-consider adding recent studies on brucellosis serology, antibody persistence, and BP26 diagnostic applications.

Figures and Tables

Improve figure quality and readability.

Ensure consistent formatting across tables (e.g., decimal places, unit usage).

Reviewer #2 (Comments for the Author):

The revised manuscript presents an important study on the dynamics of antibody changes in brucellosis patients in Gansu, China. The authors have made significant efforts to improve clarity, enhance methodological details, and strengthen data interpretation based on the revisions. The study offers valuable insights into serological monitoring and potential diagnostic markers, particularly BP26, for tracking brucellosis progression. The revised version demonstrates a marked improvement in organization, scientific rigor, and clarity. However, some aspects still require refinement for final acceptance.

Revisions Required

Question 1: Abstract: Improve Clarity and Structure

The abstract is too detailed and includes unnecessary methodological specifics. It should be concise and structured, highlighting the background, objectives, key findings, and conclusions.

Suggested improvements:

Clearly state the importance of monitoring antibody dynamics in brucellosis.

Summarize results more effectively, focusing on key findings without excessive details on methodology.

The last sentence should emphasize the clinical and public health significance rather than restating findings.

Answer 1: Thank you for your valuable comments and suggestions on our manuscript. We have carefully considered each of your points and made the following revisions in lines 23-39 of the manuscript:

1. We have streamlined the abstract to focus on the key elements: background, objectives, key findings, and conclusions.
2. We have removed unnecessary methodological details to make the abstract more concise.
3. We have clearly stated the importance of monitoring antibody dynamics in brucellosis.
4. We have summarized the results more effectively, focusing on key findings without excessive details on methodology.
5. The last sentence now emphasizes the clinical and public health significance of our findings.

Question 2: Introduction: Enhance Justification and Study Rationale

The introduction provides good background on brucellosis but could be more focused on the study rationale.

The public health importance of understanding antibody kinetics should be better integrated into the objectives.

The section on study site characteristics (Akesai Kazakh Autonomous County) is too detailed. It should be more concisely linked to the study's significance.

Answer 2: Thank you for your insightful comments and suggestions on our

manuscript. We have carefully considered each of your points and made the following revisions to the introduction section of the manuscript in lines 72-97:

1. We have refined the introduction to more clearly focus on the study rationale. “Lipopolysaccharide (LPS) is a key component of the outer membrane and serves as an important virulence factor for Brucella....the BP26 protein..... with higher specificity, offering a promising complementary biomarker” in lines 81-89.
2. We have included the importance of understanding antibody dynamics for public health in the introduction section. “Antibody dynamic analysis is central to immune monitoring, revealing the temporal characteristics and clinical significance of humoral immunity.....” in lines 72-80.
3. We have simplified the description of the characteristics of the study site (Aksai Kazak Autonomous County) and linked it to the significance of the study in lines 90-92.

Question 3: Methods: Improve Reproducibility

The sample selection criteria need more clarity. Why was 400 selected as the initial sample size? Provide justification based on statistical or epidemiological considerations.

Western blotting methodology: Provide details on quality control measures for protein purity and specificity.

Follow-up intervals: The rationale for 30-day follow-ups should be justified by referencing relevant literature.

Answer 3:

1. Thank you for your question regarding the sample selection criteria in our study. This study included 400 animal husbandry practitioners in Aksai Kazakh Autonomous County, Gansu Province, monitored by the local CDC from 2022 to 2023. This group included individuals involved in herding and other occupations with direct animal contact, identified through passive surveillance (reporting symptomatic cases) and active surveillance (screening high-risk workplaces). The sample size of 400 balanced feasibility and representativeness, ensuring coverage of the high-risk population while considering fieldwork limitations.

2. Thank you for your valuable comments and suggestions on our manuscript. We have carefully considered your request for more details on the quality control measures for protein purity and specificity in our Western blotting methodology and have made the revisions on lines 295-298 and 304-307.

3. Thank you for your review comments. We apologize for not adequately citing references to support our follow-up interval choice in the previous revision. We've now added relevant citations in the Materials and Methods section on line 276.

As per the study by Bai et al. (2023), which followed 100 acute brucellosis patients in Inner Mongolia, China, for 6 months with follow-ups at 6 weeks (42 days), 12 weeks, and 3 months post-treatment, there were significant changes in antibody titers, with a notable decrease in SAT positivity and titers ($\geq 1:200++$) at 42 days.

Given the rapid fluctuations in antibody titers during the early phase of treatment, close monitoring is critical. Considering our patient population—dispersed livestock

workers with limited access to medical resources—a 30-day follow-up interval strikes a balance between timely assessment of treatment efficacy and minimizing the burden on both patients and healthcare resources. Given the rapid fluctuations in antibody titers during early treatment, and considering that our patient group consists of livestock workers scattered across pastoral areas, making follow-up relatively difficult, we have chosen a 30-day follow-up interval. This is supported by the above literature and aims to promptly assess treatment efficacy while ensuring the smooth progress of the study.

Reference:

Bai L, Ta N, Zhao A, Muren H, Li X, Wang BC, Bagen H, Wen Y. 2023. A follow-up study of 100 patients with acute brucellosis for its prognosis and prevention. *Frontiers in Medicine* 10:1110907.

Question 4: Results: Improve Data Presentation and Interpretation

The section effectively describes antibody kinetics but needs clearer statistical validation.

Figures and tables: Figure legends should be more descriptive.

Table 1 (Western blot results) lacks statistical validation—consider adding statistical tests to compare antibody persistence trends.

The Western blot findings should discuss potential confounders, such as patient immune variability or testing conditions.

Answer 4:

1. Thank you for the valuable suggestion regarding the figure legends. We have revised the figure legends to make them more descriptive and informative.

Thank you for your comment on the lack of statistical validation in Table 1 (Western blot results). We appreciate your suggestion to include statistical tests to compare antibody persistence trends. However, we need to clarify that Table 1 aims to clearly present the qualitative results of Western Blot testing (such as antibody positive/negative status and reactivity with BP26 and LPS), not to quantitatively compare antibody titers dynamics. Consequently, for the following reasons, conventional statistical tests cannot be directly applied to this table:

- ① Inconsistent patient follow-up times misalign time-series data.
- ② Significant variability in humoral immune responses to *Brucella* antigens (e.g., LPS, BP26) among patients makes it difficult to draw generalizable conclusions through between-group mean or proportion comparisons.
- ③ Small sample size may introduce statistical bias (e.g., risk of Type II errors).

2. Thank you for your insightful comment regarding the potential confounders in the Western blot findings. We agree that factors such as patient immune variability and testing conditions can influence the results. In response, we have added a discussion on these potential confounders in the revised manuscript in lines 212 to 224.

Question 5: Discussion: Strengthen Interpretation and Clinical Implications

The discussion is comprehensive but somewhat redundant. It should be more structured, focusing on:

Key findings and their implications for brucellosis diagnosis and treatment.

Comparison with previous literature-how do these findings align with or contradict other serological studies?

Potential applications of BP26 protein detection-can it be used for early diagnosis or vaccine development?

The implications for brucellosis surveillance and control strategies should be elaborated.

Answer 5: Thank you for your insightful feedback. We appreciate your comment regarding the structure and redundancy in the Discussion section. In response, we have revised the section to ensure a more streamlined and well-structured presentation of the key findings, Comparison with previous literature, Potential applications and The implications for brucellosis surveillance and control strategies.

1. For the key findings and their implications for brucellosis diagnosis and treatment. We explain this in lines 171-175 of the manuscript. “ Acute brucellosis patients displayed a distinct three-phase antibody titer pattern: a rapid rise basis for optimizing diagnosis and treatment routes and monitoring strategies.”

2. For the comparison with previous literature. We explain this in lines 176-195 of the manuscript. “The cases of brucellosis have been on the rise in China in recent years... ..Brucella can survive in cells for a long time, potentially by suppressing immunity, autophagy, and apoptosis through various mechanisms.”

3. For the potential applications of BP26 protein detection. We explain this in lines 201-211 of the manuscript. “In this study, Western blot analysis revealed that BP26 antibodies persist longer than LPS antibodies... .. BP26 protein early diagnosis, detection of chronic brucellosis patients, and vaccine development”

4. For the implications for brucellosis surveillance and control strategies. We explain this in lines 225-234 of the manuscript. “Currently, brucellosis surveillance primarily relies on passive surveillanceenhanced surveillance should be complemented by community-level education on brucellosis transmission and prevention.”

Question 6: Limitations: Acknowledge Study Constraints More Explicitly

While limitations are mentioned, the following should be explicitly acknowledged:

Small sample size (n=21 positive cases), limiting generalizability.

Non-uniform follow-up intervals, which may introduce variability in antibody kinetics assessment.

No pathogen isolation or species-level *Brucella* identification, which limits broader applicability.

Answer 6: Thank you for your valuable comments and suggestions on our manuscript. We have carefully considered each of your points and made the revisions in lines 234-238 of the manuscript. These revisions aim to provide a more transparent and comprehensive understanding of our study's limitations, ensuring that readers can appropriately interpret our findings within these constraints.

Question 7: Grammar and Style Improvements

There are grammatical errors and awkward phrasing that need refinement.

Standardized terminology should be consistently used (e.g., "Brucella seropositivity" instead of "Brucella antibody-positive cases").

Answer 7: Thank you for your careful review of our manuscript. In response to your suggestions for grammar and style improvements. We have corrected the grammatical errors and awkward phrasing in the article to ensure that the language is clearer and more accurate.

Question 8: References

Some references are outdated-consider adding recent studies on brucellosis serology, antibody persistence, and BP26 diagnostic applications.

Answer 8: We appreciate the reviewer's valuable suggestions regarding our work. Regarding the update of the literature, we have reviewed and added several important recent studies on *Brucella* serology, antibody persistence, and the diagnostic application of BP26, see references (22), (23), (25), (26). These studies provide valuable additions and updates to some of the discussions in our paper. We have made corresponding revisions to ensure the timeliness and comprehensiveness of the literature. If the reviewers have any other recommended references, we would be happy to further review and incorporate them.

22. Wu Q, Yuan L, Guo X, Sun M, Yao M, Yin D. 2023. Study on antigenic protein Omp2b in combination with Omp31 and BP26 for serological detection of human brucellosis. *Journal of Microbiological Methods* 205:106663.

23. Nagalingam M, Basheer TJ, Balamurugan V, Shome R, Kumari SS, Reddy GBM, Shome BR, Rahman H, Roy P, Kingston JJ, Gandham RK. 2021. Comparative evaluation of the immunodominant proteins of *Brucella abortus* for the diagnosis of cattle brucellosis. *Veterinary World* 14:803-812.

25. Xie Y, Lin S, Guo L, Qi X, Zhao S, Pei Q, Chen Y, Wu Q, Wang Y, Yao M, Yin D. 2024. Development and evaluation of the recombinant BP26 protein-based C-ELISA for human brucellosis diagnosis. *Front Microbiol* 15:1516915.

26. Guo X, Sun M, Guo Y, Wu Y, Yan X, Liu M, Li J, Sun X, Fan X, Zhang H, Sun S, Wang J, Yin D. 2024. Production and evaluation of anti-BP26 monoclonal antibodies for the serological detection of animal brucellosis. *Frontiers in Veterinary Science* 11.

Question 9: Figures and Tables

Improve figure quality and readability.

Answer 9: Thank you to the reviewer for the valuable suggestions regarding the quality and readability of the figures. Based on the reviewer's comments, we have raised the resolution of all figures to ensure they are displayed more clearly across different devices, and adjusted the fonts and labels in all figures to ensure the font size is suitable for both print and online viewing, and have made sure that axis labels, legends, and titles are legible.

Re: Spectrum02862-24R3 (**Follow-up of antibody changes in brucellosis patients in Gansu, China**)

Dear Prof. Xin Wang:

Your manuscript has been accepted, and I am forwarding it to the ASM production staff for publication. Your paper will first be checked to make sure all elements meet the technical requirements. ASM staff will contact you if anything needs to be revised before copyediting and production can begin. Otherwise, you will be notified when your proofs are ready to be viewed.

Sincerely,
Siu-Kei Chow
Editor
Microbiology Spectrum